# Iso-Dream: Isolating and Leveraging Noncontrollable Visual Dynamics in World Models

**Minting Pan**[*]     **Xiangming Zhu**[*]     **Yunbo Wang**[†]     **Xiaokang Yang**
MoE Key Lab of Artificial Intelligence, AI Institute, Shanghai Jiao Tong University
{panmt53, xmzhu76, yunbow, xkyang}@sjtu.edu.cn

## Abstract

World models learn the consequences of actions in vision-based interactive systems. However, in practical scenarios such as autonomous driving, there commonly exists noncontrollable dynamics independent of the action signals, making it difficult to learn effective world models. To tackle this problem, we present a novel reinforcement learning approach named Iso-Dream, which improves the Dream-to-Control framework [22] in two aspects. First, by optimizing the *inverse dynamics*, we encourage the world model to learn controllable and noncontrollable sources of spatiotemporal changes on isolated state transition branches. Second, we optimize the behavior of the agent on the decoupled latent imaginations of the world model. Specifically, to estimate state values, we roll-out the *noncontrollable states* into the future and associate them with the current *controllable state*. In this way, the isolation of dynamics sources can greatly benefit long-horizon decision-making of the agent, such as a self-driving car that can avoid potential risks by anticipating the movement of other vehicles. Experiments show that Iso-Dream is effective in decoupling the mixed dynamics and remarkably outperforms existing approaches in a wide range of visual control and prediction domains.

## 1 Introduction

Humans can infer and predict real-world dynamics by simply observing and interacting with the environment. Inspired by this, many cutting-edge AI agents use self-supervised learning [37, 20, 12] or reinforcement learning [38, 22, 41] techniques to acquire knowledge from their surroundings. Among them, world models [20] have received widespread attention in the field of robot visual control, and led the recent progress in model-based reinforcement learning (MBRL) [22, 41, 24, 29]. A typical approach [22] is to use the trajectories of observations and control signals collected by an RL agent to learn a differentiable simulator of the environment, namely the world model, and then update the RL agent by optimizing the behaviors on the latent *imaginations* of the world model.

However, since the observation sequence is high-dimensional, non-stationary, and often driven by multiple sources of physical dynamics, how to learn effective world models in complex visual scenes remains an open problem. In realistic scenarios such as autonomous driving, we can generally divide spatiotemporal dynamics in the system into controllable parts that perfectly respond to action signals, and parts beyond the control of the agent, such as the movement of other vehicles and other external changes. The isolation of controllable and noncontrollable states can improve MBRL in two aspects:

- Modular representation improves the generalization of the agent to non-stationary environments with noises, such as the time-varying background in our modified DeepMind Control Suite.

---

[*]Equal contribution.
[†]Corresponding author: Yunbo Wang.
Code available at `https://github.com/panmt/Iso-Dream`

36th Conference on Neural Information Processing Systems (NeurIPS 2022).

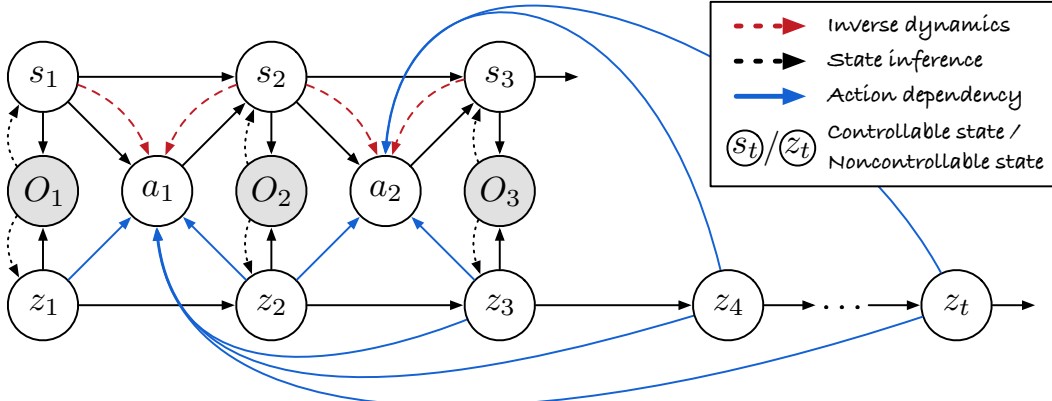

Figure 1: Probabilistic graph of Iso-Dream. It learns to decouple complex visual dynamics into controllable states ($s_t$) and noncontrollable states ($z_t$) by optimizing the inverse dynamics (Red dashed arrows). On top of the disentangled states, it performs model-based reinforcement learning by explicitly considering the predicted noncontrollable component of future dynamics (Blue arrows).

- More importantly, it improves long-horizon RL tasks that can greatly benefit from decisions based on predictions of future noncontrollable dynamics. For example, in autonomous driving, potential risks can be better avoided by predicting the movement of other vehicles.

We present Iso-Dream, a novel MBRL framework that learns to decouple and leverage the controllable and noncontrollable state transitions. Accordingly, it improves the original Dreamer [22] from two perspectives: (i) *a new form of world model representations* and (ii) *a new actor-critic algorithm to derive the behavior from the world model*. As shown in Figure 1, the foundation of decoupling the world model is to separate the mixed latent states into an action-conditioned branch and an action-free branch, which can individually transit different sources of visual dynamics. The components are jointly trained to maximize the variational lower bounds. To further isolate the controllable states, the action-conditioned branch is also optimized with inverse dynamics, that is, to reason about the actions that have driven the state transitions between adjacent time steps.

Another contribution of Iso-Dream is to find that disentangling physical dynamics can greatly benefit the downstream decision-making tasks by more accurately foreseeing the inherent changes in the environment. Intuitively, humans can decide how to interact with the environment at each moment based on their anticipation of future changes in the surroundings. To make more forward-looking decisions, as shown by the blue arrows in Figure 1, the policy network integrates the current controllable state and multiple steps of predicted noncontrollable states through an attention mechanism. It enables the agent to thoroughly consider possible future interactions with the environment.

We evaluate Iso-Dream in the following domains: The modified DeepMind Control Suite with noisy video background; The CARLA autonomous driving environment in which other vehicles can be naturally viewed as noncontrollable components; The real-world BAIR robot dataset and the RoboNet dataset that are helpful to validate the effectiveness of the world model for disentanglement. On all benchmarks, Iso-Dream remarkably outperforms the existing approaches by large margins.

## 2 Related Work

**Action-conditioned video prediction.** A straightforward deep learning solution to visual control problems is to learn action-conditioned video prediction models [37, 14, 8, 52] and then perform Monte-Carlo importance sampling and optimization algorithms, such as the *cross-entropy methods*, over available behaviors [15, 12, 28]. Hot topics in video prediction mainly includes long-term and high-fidelity future frames generation [43, 42, 50, 5, 51, 49, 53, 40, 39, 35, 55, 27, 2], dynamics uncertainty modeling [1, 10, 47, 30, 7, 16, 54], object-centric scene decomposition [46, 26, 18, 57, 3], and space-time disentanglement [48, 26, 19, 6]. The corresponding technical improvements mainly involve the use of more effective neural architectures, novel probabilistic modeling methods, and specific forms of video representation. The disentanglement methods are closely related to the world

model in Iso-Dream. They commonly separate visual dynamics into content and motion vectors, or long-term and short-term states. In contrast, Iso-Dream is designed to learns a decoupled world model based on controllability, which contributes more to the downstream behavior learning process.

**Visual MBRL.** In visual control tasks, the agents have to learn the action policy directly from high-dimensional observations. They can be roughly grouped into two categories, that is, model-free methods [33, 56, 31, 32, 25] and model-based methods [15, 38, 20, 23, 22, 29, 41, 58, 4]. Among them, the MBRL approaches explicitly model the state transitions and generally yield higher sample efficiency than the model-free methods. Ha and Schmidhuber [20] proposed the World Models that first learn compressed latent states of the environment in a self-supervised manner, and then train the agent on the latent states generated by the world model. Following the two-stage training procedure, PlaNet [23] uses an action-conditioned, recurrent state-space model (RSSM) as the world model, and optimizes the action policy on the recurrent states with the cross-entropy methods. In Dreamer [22] and DreamerV2 [24], agents learn behaviors by optimizing the expected values over the predicted latent states in RSSM. InfoPower [4] prioritizes functional-related information from visual observations to obtain a more robust representation for MBRL. Notably, Iso-Dream is very different from InfoPower in two aspects. First, we explicitly model the state transitions of controllable and noncontrollable dynamics, so that it is possible to choose whether to take the noncontrollable states into behavior learning according to the prior knowledge of a specific domain. Second, we propose a new behavior learning method that greatly benefits from the decoupled world model, so that we can preview possible future states of noncontrollable patterns before making decisions at this moment.

## 3 Method

In this section, we first present basic assumptions and the general framework of Iso-Dream for decoupling and leveraging controllable and noncontrollable dynamics for visual control (Section 3.1). For representation learning, we introduce the three-branch world model and its training objectives of inverse dynamics (Section 3.2). For behavior learning, we present an actor-critic method that is trained on the imaginations of the decoupled world model latent states, so that the agent may consider possible future states of noncontrollable dynamics (Section 3.3). Finally, we discuss how Iso-Dream is deployed to interact with the environment (Section 3.4).

### 3.1 Basic Assumptions of Iso-Dream

As shown in Figure 1, when the agent receives a sequence of visual observations $o_{1:T}$, the underlying spatiotemporal dynamics can be defined as $u_{1:T}$. Our goal is to understand the inner relationships of different dynamics by decoupling $u_{1:T}$ into controllable latent states $s_{1:T}$ and noncontrollable latent states $z_{1:T}$ that vary in spacetime, such that:

$$u_{1:T} \sim (s, z)_{1:T}, \quad s_{t+1} \sim p(s_{t+1} \mid s_t, a_t), \quad z_{t+1} \sim p(z_{t+1} \mid z_t), \tag{1}$$

where $a_t$ is the action signal. To achieve long-term prediction, we isolate $s_t$ and $z_t$ to each other and model their state transitions of $p(s_{t+1} \mid s_t, a_t)$ and $p(z_{t+1} \mid z_t)$ respectively.

According to our prior knowledge of the environment, we can optionally choose whether to roll out the noncontrollable states and consider them during behavior learning. For tasks where the noncontrollable components can be viewed as time-varying noise, we simply derive the action policy by $a_t \sim \pi(a_t \mid s_t)$. The isolation of controllable states improves the generalization of the agent to non-stationary systems. For tasks like autonomous driving, the behaviors are derived by

$$a_t \sim \pi(a_t \mid s_t, z_{t:t+\tau}), \tag{2}$$

where we calculate the relationships between $s_t$ and the imagined noncontrollable states over time horizon $\tau$. It assumes that, in specific long-horizon tasks, the agent can greatly benefit from predicting the consequences of external noncontrollable forces.

### 3.2 Representation Learning of Controllable and Noncontrollable Dynamics

Inspired by previous approaches [36, 17] showing that modular structures are effective for disentanglement learning, we leverage a three-branch architecture to decouple $u_t$ into controllable dynamics state $s_t$, noncontrollable dynamics state $z_t$, and time-invariant representation of the background. As

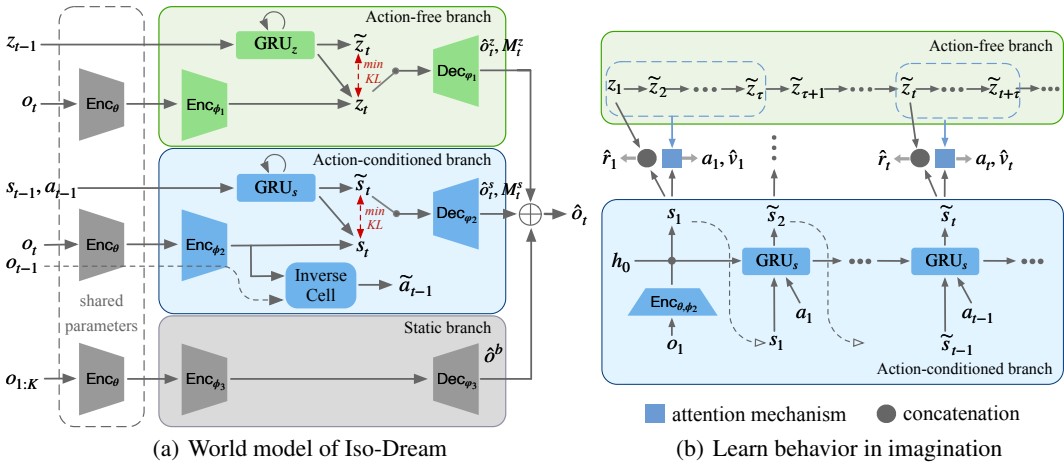

| (a) World model of Iso-Dream | (b) Learn behavior in imagination |

Figure 2: The overall architecture of the world model and the behavior learning algorithm in Iso-Dream. (a) World model with three branches to explicitly disentangle controllable, noncontrollable, and static components from visual data, where the action-conditioned branch learns controllable state transitions by modeling inverse dynamics. (b) The agent optimizes the behaviors in imaginations of the world model through a future state attention mechanism.

shown in Figure 2(a), the action-conditioned branch models $p(s_{t+1} \mid s_t, a_t)$. It follows the RSSM architecture from PlaNet [23] to use a recurrent neural network $\text{GRU}_s(\cdot)$, the deterministic hidden state $h_t$, and the stochastic state $s_t$ to form the transition model, where the GRU keeps the historical information of the controllable dynamics. The action-free branch models $p(z_{t+1} \mid z_t)$ with similar network structures. The transition models with separate parameters can be written as follows:

$$p(\tilde{s}_t \mid s_{<t}, a_{<t}) = p(\tilde{s}_t \mid h_t), \quad \text{where} \ \ h_t = \text{GRU}_s(h_{t-1}, s_{t-1}, a_{t-1}),$$
$$p(\tilde{z}_t \mid z_{<t}) = p(\tilde{z}_t \mid h'_t), \quad \text{where} \ \ h'_t = \text{GRU}_z(h'_{t-1}, z_{t-1}). \tag{3}$$

We here use $\tilde{s}_t$ and $\tilde{z}_t$ to denote the prior representations. We optimize the transition models with posterior representations that are derived from $s_t \sim q(s_t \mid h_t, o_t)$ and $z_t \sim q(z_t \mid h'_t, o_t)$. We learn the posteriors from the observation at current time step $o_t \in \mathbb{R}^{3 \times H \times W}$ by a shared encoder $\text{Enc}_\theta$ and subsequent branch-specific encoders $\text{Enc}_{\phi_1}$ and $\text{Enc}_{\phi_2}$.

To enhance the disentanglement representation learning corresponding to the control signals, we introduce the training objective of *inverse dynamics*. Accordingly, we design an Inverse Cell of a 2-layer MLP to infer the actions that lead to certain transitions of the controllable states:

$$\text{Inverse dynamics:} \quad \tilde{a}_{t-1} = \text{MLP}(s_{t-1}, s_t), \tag{4}$$

where the inputs are the posterior representations in the action-conditioned branch. By learning to regress the true behavior $a_{t-1}$, the Inverse Cell facilitates the action-conditioned branch to isolate the representation of the controllable dynamics. To avoid the training collapse where the action-conditioned branch captures most of the useful information, while the action-free branch learns almost nothing, in the process of image reconstruction, we respectively use the prior state $\tilde{s}_t$ and the posterior state $z_t$ to generate the controllable visual component $\hat{o}^s_t \in \mathbb{R}^{3 \times H \times W}$ with mask $M^s_t \in \mathbb{R}^{1 \times H \times W}$ and the noncontrollable component $\hat{o}^z_t \in \mathbb{R}^{3 \times H \times W}$ with $M^z_t \in \mathbb{R}^{1 \times H \times W}$. By further integrating the time-invariant information extracted from the first $K$ frames, we have

$$\hat{o}_t = M^s_t \odot \hat{o}^s_t + M^z_t \odot \hat{o}^z_t + (1 - M^s_t - M^z_t) \odot \hat{o}^b, \quad \text{where} \ \ \hat{o}^b = \text{Dec}_{\varphi_3}(\text{Enc}_{\theta, \phi_3}(o_{1:K})). \tag{5}$$

For reward modeling, we have two options with the action-free branch. In one case, the noncontrollable dynamics can be considered as noises that are not related to the task, and therefore $z_t$ is no longer useful during imagination. In other words, the policy and the predicted reward are only related to the controllable states. In the other case, future noncontrollable states would affect how the agent makes decisions, and we consider the action-free components during behavior learning. For this, we learn alternative reward models $p(r_t \mid s_t)$ or $p(r_t \mid s_t, z_t)$ in forms of MLPs.

**Algorithm 1:** Iso-Dream (Highlight: Our modifications to behavior learning & policy deployment)

1   **Hyperparameters:** $L$: Imagination horizon; $\tau$: Window size for future state attention
2   Initialize the replay buffer $\mathcal{B}$ with random episodes.
3   **while** *not converged* **do**
4     **for** *update step* $c = 1 \ldots C$ **do**
5       Draw data sequences $\{(o_t, a_t, r_t)\}_{t=1}^{T} \sim \mathcal{B}$.
6       `// Representation learning`
7       Compute world model loss using Eq. (6) and update model parameters.
8       `// Behavior learning`
9       Roll-out the noncontrollable states $\{\tilde{z}_i\}_{i=t+1}^{t+L+\tau}$ from $z_t$ through the action-free branch alone.
10       **for** *time step* $j = i \ldots i + L$ **do**
11         Compute latent state $e_j \sim \texttt{Attention}(\tilde{s}_j, \tilde{z}_{j:j+\tau})$ using Eq. (7).
12         Imagine an action $a_j \sim \pi(a_j | e_j)$.
13         Predict the next controllable state $\tilde{s}_{j+1} \sim p(\tilde{s}_j, a_j)$ using the action-conditioned branch alone.
14       **end**
15       Update the policy and value models in Eq. (8) using estimated rewards and values.
16     **end**
17     `// Environment interaction`
18     $o_1 \leftarrow \texttt{env.reset()}$
19     **for** *time step* $t = 1 \ldots T$ **do**
20       Calculate the posterior representation $s_t \sim q\left(s_t \mid h_t, o_t\right), z_t \sim q\left(z_t \mid h_t', o_t\right)$.
21       Roll-out the noncontrollable states $\tilde{z}_{t+1:t+\tau}$ from $z_t$ through the action-free branch alone.
22       Generate $a_t \sim \pi(a_t \mid s_t, z_t, \tilde{z}_{t+1:t+\tau})$ using future state attention in Eq. (7).
23       $r_t, o_{t+1} \leftarrow \texttt{env.step}(a_t)$
24     **end**
25     Add experience to the replay buffer $\mathcal{B} \leftarrow \mathcal{B} \cup \{(o_t, a_t, r_t)_{t=1}^{T}\}$.
26   **end**

For a sequence of $(o_t, a_t, r_t)_{t=1}^{T}$ sampled from the replay buffer during training, the world model can be optimized using the following loss functions, where $\alpha$, $\beta_1$, and $\beta_2$ are hyper-parameters:

$$\mathcal{L} = \mathrm{E}\{\sum_{t=1}^{T} \underbrace{-\ln p(o_t \mid h_t, s_t, h_t', z_t)}_{\text{image log loss}} \underbrace{-\ln p(r_t \mid h_t, s_t, h_t', z_t)}_{\text{reward log loss}} \underbrace{-\ln p(\gamma_t \mid h_t, s_t, h_t', z_t)}_{\text{discount log loss}}$$
$$+ \underbrace{\alpha \ell_2(a_t, \tilde{a}_t)}_{\text{action loss}} + \underbrace{\beta_1 \mathrm{KL}[q(s_t \mid h_t, o_t) \mid p(s_t \mid h_t)] + \beta_2 \mathrm{KL}[q(z_t \mid h_t', o_t) \mid p(z_t \mid h_t')]}_{\text{KL divergence}}\}. \quad (6)$$

The world model training approach can be partly customized for different environments. In situations where noncontrollable states are indeed involved in behavior learning, minimizing the ELBO objective can maintain the semantics of $\tilde{z}_t$. Otherwise, if the action-free features are only used to prevent noisy distractions from affecting the training process of Iso-Dream, rather than being used for behavior learning, we can simply train the action-free branch with the reconstruction loss alone.

### 3.3   Behavior Learning in Decoupled Imaginations

Thanks to the decoupled world model, we can optimize the agent behaviors to adaptively consider the relations between available actions and possible future states of the noncontrollable dynamics. A practical example is autonomous driving, where the motion of other vehicles can be naturally viewed as noncontrollable but predictable components. As shown in Figure 2(b), we here propose an improved actor-critic learning algorithm that *1) allows the action-free branch to foresee the future ahead of the action-conditioned branch, and 2) exploits the predicted future information of noncontrollable dynamics to make more forward-looking decisions.*

Suppose we are making decisions at time step $t$ in the imagination period. A straightforward solution from the original Dreamer method is to learn an action model and a value model based on the isolated controllable state $\tilde{s}_t \in \mathbb{R}^{1 \times d}$. However, we notice that by employing an attention mechanism, we can explicitly calculate its relations to a sequence of future noncontrollable states $\tilde{z}_{t:t+\tau} \in \mathbb{R}^{\tau \times d}$, where

$\tau$ is the length of a sliding window from now on.

$$\text{Future state attention:} \quad e_t = \text{softmax}(\tilde{s}_t \ \tilde{z}_{t:t+\tau}^T) \ \tilde{z}_{t:t+\tau} + \tilde{s}_t. \tag{7}$$

In this way, $\tilde{s}_t$ evolves to a more "*visionary*" representation $e_t \in \mathbb{R}^{1 \times d}$. We update the action model and the value model in Dreamer [22] as follows:

$$\text{Action model:} \quad a_t \sim \pi(a_t \mid e_t), \quad \text{Value model:} \quad v_\xi(e_t) \approx \mathbb{E}_{\pi(\cdot \mid e_t)} \sum_{k=t}^{t+L} \gamma^{k-t} r_k, \tag{8}$$

where $L$ is the imagination time horizon. As shown in Alg. 1, during imagination, we first use the action-free transition model to obtain sequences of noncontrollable states of length $L + \tau$, denoted by $\{\tilde{z}_i\}_{i=t}^{i+L+\tau}$. At each time step in the imagination period, the agent draws an action $a_j$ from the visionary state $e_j$, which is derived from Eq. (7). The action-conditioned branch uses the action $a_j$ in latent imagination and predicts the next controllable state $s_{j+1}$. We follow DreamerV2 [24] to train the action model to maximize the $\lambda$-return [44], and train the value model to regress the $\lambda$-return[3].

### 3.4 Policy Deployment by Rolling-out Noncontrollable Dynamics

As discussed above, in the cases that noncontrollable dynamics are irrelevant to the control task, when interacting with the environment, we only use the state of controllable dynamics to generate the policy at each time step $t$. However, for the situation where noncontrollable dynamics should be closely related to the behavior of the agent, as shown in Lines 21-22 in Alg. 1, the action-free branch consecutively predicts the next $\tau - 1$ noncontrollable states $\tilde{z}_{t+1:t+\tau}$ starting from the current posterior state $z_t$. Similar to Eq. (7) in the process of behavior learning, we here use the learned future state attention network to adaptively integrate $s_t$, $z_t$ and $\tilde{z}_{t+1:t+\tau}$. Based on the integrated feature $e_t$, the Iso-Dream agent draws $a_t$ from the action model to interact with the environment.

## 4 Experiments

### 4.1 Experimental Setup

**Benchmarks.** We quantitatively and qualitatively evaluate Iso-Dream on two reinforcement learning environments, *i.e.*, DeepMind Control Suite [45] and CARLA [11], and two real-world datasets for action-conditioned video prediction, *i.e.*, BAIR robot pushing [13] and RoboNet [9]. The video prediction experiments can provide more intuitive visualizations of disentanglement learning.

**Compared methods.** For the visual control tasks, we compare our approach with five baselines, including both model-based and model-free methods, *i.e.*, DreamerV2 [24], CURL [33], SVEA [25], SAC [21], and DBC [58]. For action-conditioned video prediction, we mainly compare our decoupled world model with three approaches, *i.e.*, SVG [10], SA-ConvLSTM [34] and PhyDNet [19].

### 4.2 DeepMind Control Suite

**Implementation details.** In order to verify the enhancement of Iso-Dream by disentangling different components under complex visual dynamics, we evaluate Iso-Dream on environments from DMC Generalization Benchmark. Instead of training on original DeepMind Control Suite environments, agents are trained and tested both with natural video backgrounds (*i.e.* video_easy environments). In this environment, since the background is randomly replaced by a real-world video, the noncontrollable motion of the background can affect the procedure of dynamics learning and behavior learning of agents. Therefore, to obtain a better decision policy and avoid the disruption from noisy backgrounds, the agent may decouple noncontrollable representation (*i.e.*, dynamic background) and controllable representation in spacetime, and only use controllable representation for control. To this end, we simply train the action-free branch with only reconstruction loss and discard it in imagination and policy deployment. We evaluate our model with baselines in 4 tasks from four different domains. The number of environment steps is limited to 500k.

---

[3]Details of the loss functions can be found in Eq. (5-6) in the paper of DreamerV2 [24] .

Table 1: Performance of visual control tasks in the DMC Suite. The agents are trained and evaluated in environments with `video_easy` dynamic background. We report the mean and std of final performance over 3 seeds and 5 trajectories. *We use a different setup from that in the paper of DBC.

| TASK | SVEA | CURL | DBC* | DREAMERV2 | ISO-DREAM |
|------|------|------|------|-----------|-----------|
| WALKER WALK | $826 \pm 65$ | $443 \pm 206$ | $32 \pm 7$ | $655 \pm 47$ | $\mathbf{911 \pm 50}$ |
| CHEETAH RUN | $178 \pm 64$ | $269 \pm 24$ | $15 \pm 5$ | $475 \pm 159$ | $\mathbf{659 \pm 62}$ |
| FINGER SPIN | $562 \pm 22$ | $280 \pm 50$ | $1 \pm 2$ | $755 \pm 92$ | $\mathbf{800 \pm 59}$ |
| HOPPER STAND | $6 \pm 8$ | $451 \pm 250$ | $5 \pm 9$ | $260 \pm 366$ | $\mathbf{746 \pm 312}$ |

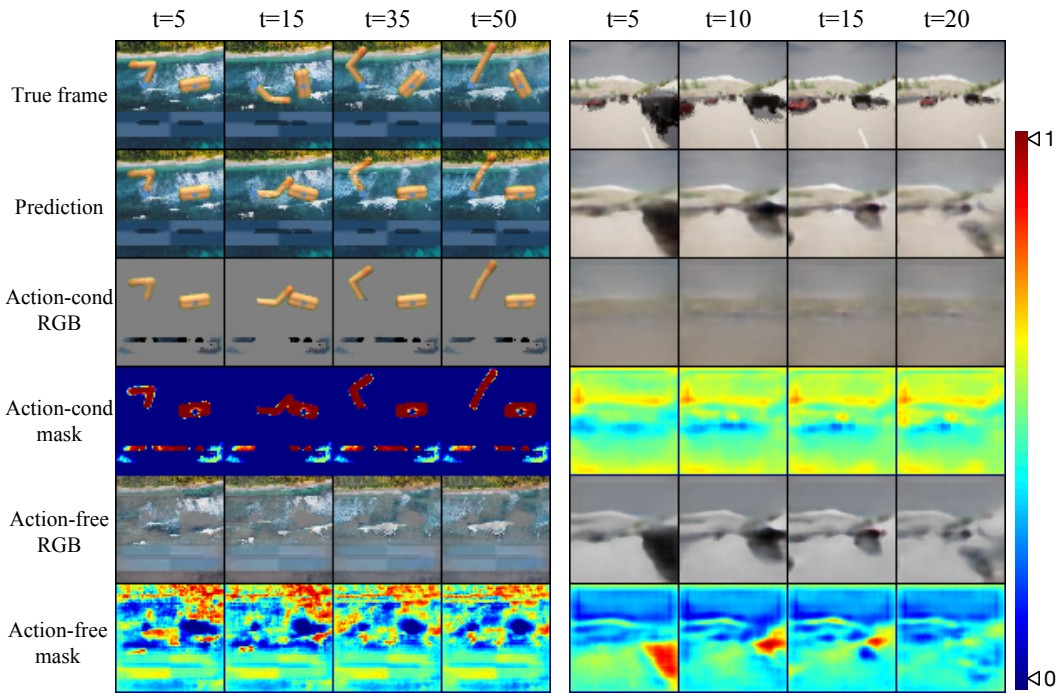

Figure 3: Video prediction results on the DMC (**left**) and CARLA (**right**) benchmarks of Iso-Dream. For each sequence, we use the first 5 images as context frames. Iso-Dream successfully disentangles controllable and noncontrollable components.

**Quantitative results.**    To evaluate the performance, we train and test the agents in environments with video backgrounds. As shown in Table 1, Iso-Dream exceeds the performance of DreamerV2 and other baselines in all tasks, indicating that the three-branch structure can effectively learn task-related visual representations and alleviate complex background interference in visual data.

**Qualitative results.**    We leverage Iso-Dream to complete video prediction tasks in `video_easy` environments. The sequence of frames and actions are randomly collected during test episodes. The first 5 frames are given to the model and the next 45 frames are predicted only based on action inputs. To show the qualitative results, we visualize the masks and visual decoupled components from the action-conditioned and action-free branches. The overall visualization is shown in Figure 3(left). From this prediction result, we can find that Iso-Dream has the ability to predict long-term sequence and disentangle controllable and noncontrollable dynamics from images in `video_easy` environments. As shown in the third and fourth row of action-conditioned branch output in Figure 3, the controllable representation has been successfully isolated and matches its mask. Besides, in this visualization, the action-free component in this background video is the motion of sea waves, which is captured by the fifth and sixth row of action-free branch outputs.

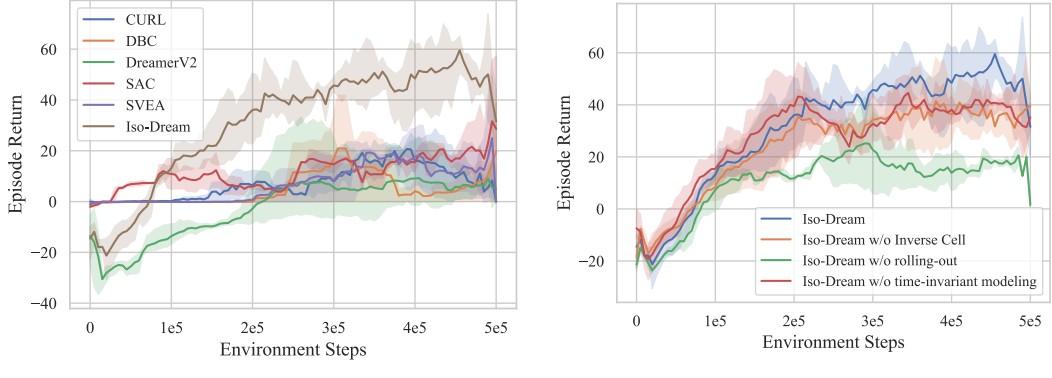

(a) Comparison with the state-of-the-arts.     (b) Ablation study of Iso-Dream.

Figure 4: Performance with 3 seeds on the CARLA driving task. **(a)** Comparison of existing methods, in which Iso-Dream outperforms DreamerV2 by a large margin. **(b)** Ablation studies that can show the respective impact of optimizing the inverse dynamics (orange), rolling out noncontrollable states (green), and modeling the time-invariant information with a separate network branch (red).

## 4.3 CARLA Autonomous Driving Environment

**Implementation details.** In the autonomous driving task, We use a camera with 60 degree view on the roof of the ego-vehicle, which obtains images of $64 \times 64$ pixels. Following the setting in the DBC [58], in order to encourage highway progression and penalise collisions, the reward is formulated as: $r_t = v_{ego}^T \hat{u}_h \cdot \Delta t - \xi_1 \cdot \mathbb{I} - \xi_2 \cdot |steer|$, where $v_{ego}$ is the velocity vector of the ego-vehicle, projected onto the highway's unit vector $\hat{u}_h$, and multiplied by time discretization $\Delta t = 0.05$ to measure highway progression in meters. Impulse $\mathbb{I} \in \mathbb{R}^+$ is caused by collisions, and a steering penalty $steer \in [-1, 1]$ facilitates lane-keeping. The hyper-parameters $\xi_1$ and $\xi_2$ are set to $10^{-4}$ and 1, respectively. We use $\beta_1 = 1$, $\beta_2 = 1$ and $\alpha = 1$ in Eq. (6) and $\tau = 5$ in Eq. (7).

**Quantitative results.** As shown in Figure 4(a), Iso-Dream has significant advantages compared to other baselines and outperforms DreamerV2 by a large margin. Furthermore, we conduct ablation studies to confirm the validity of inverse dynamics and the rolling-out strategy of noncontrollable states. Figure 4(b) shows that the performance drops when Inverse Cell is removed, indicating the importance of modeling inverse dynamics to isolate controllable and noncontrollable components from the whole dynamics. In order to verify the effectiveness of the proposed attention mechanism, we conduct experiments to evaluate Iso-Dream where policy networks directly concatenate the current controllable state and the noncontrollable state as input. Comparing the blue curve and green curve, we observe that rolling-out noncontrollable states in the action-free branch can significantly improve the agent's decision-making results. The red curve shows that the performance of Iso-Dream degrades by about $15\%$ in the absence of a separate network branch that captures the static information.

**Qualitative results.** Reconstruction results of predictions in CARLA environment are shown in Figure 3(right column). In CARLA, we observe that the agent actions potentially affect all pixel values in the observation, as the camera on the main car (*i.e.*, the agent) moves. Therefore, we view the visual dynamics of other vehicles as a combination of controllable and noncontrollable states. Accordingly, our model can determine which component is dominant by learning attention masks (values between 0 and 1) across the action-conditioned and action-free branches. The "action-free masks" present hot spots around other vehicles, while the attention values in corresponding areas on the "action-cond masks" are still greater than 0. The agent can avoid collisions by rolling-out noncontrollable components to preview possible future states of other vehicles. We include more showcases with different numbers of vehicles in the supplementary materials.

## 4.4 BAIR & RoboNet for Action-Conditioned Video Prediction

**Implementation details.** In order to evaluate the effectiveness of our world model in a more complex environment, we test the video prediction ability of the proposed structure on BAIR and

Table 2: Video prediction results on BAIR and RoboNet datasets with bouncing balls. We use the first 2 frames as input to predict the next 28 frames on BAIR and the next 18 frames on RoboNet.

| MODEL | BAIR | | ROBONET | |
| --- | --- | --- | --- | --- |
| | PSNR ↑ | SSIM ↑ | PSNR ↑ | SSIM ↑ |
| SVG [10] | 18.12 | 0.712 | 19.86 | 0.708 |
| SA-CONVLSTM [34] | 18.28 | 0.677 | 19.30 | 0.638 |
| PHYDNET [19] | 18.91 | 0.743 | 20.89 | 0.727 |
| ISO-DREAM | **19.51** | **0.768** | **21.71** | **0.769** |

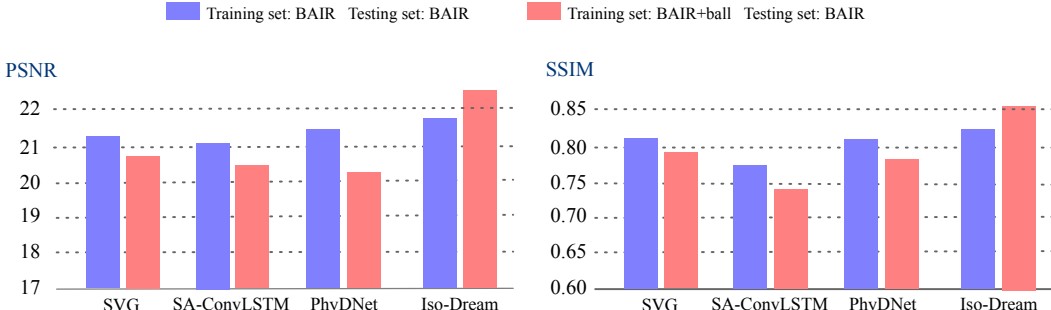

Figure 5: The results of models trained on BAIR (blue) and BAIR + bouncing balls (red), and tested on BAIR. We use the first 2 frames as input to predict the next 18 frames. The horizontal axis represents the different models, and the vertical axes represent test results of PSNR and SSIM.

RoboNet dataset. Moreover, we add predictable visual dynamics unrelated to the control signals to the raw observations, *i.e.*, bouncing balls of the same size and speed. In the training phase, we train the model to predict 10 frames into the future from 2 observations. For testing, we use the first 2 frames as input to predict the next 28 frames in the BAIR dataset, and the next 18 frames in the RoboNet dataset. All inputs for training and testing are resized to $64 \times 64$. Considering the simplicity and predictability of bouncing balls, in the action-free branch, we use a similar structure as in the DMC experiment. Moreover, we replace the GRU cell with two layers of ST-LSTM unit [51] in both branches. The optimization objective consists of image reconstruction loss and action reconstruction loss of Inverse Cell. SSIM and PSNR are adopted as evaluation metrics.

**Quantitative results.** Table 2 gives the quantitative results on BAIR and RoboNet datasets with bouncing balls in the training and testing phase. Compared with other models, Iso-Dream shows the competitive performance in two datasets. For PSNR, Iso-Dream improves SVG by 7.7% in BAIR and 9.3% in RoboNet. Compared with PhyDNet, which also disentangles features in two branches, Iso-Dream achieves better performance in both PSNR and SSIM. It shows that our Iso-Dream has a stronger ability of disentanglement learning to achieve long-term prediction. Moreover, Figure 5 shows an interesting result of the different training sets (*i.e.*, BAIR, BAIR+bouncing balls) and the same testing set (*i.e.*, BAIR). Iso-Dream is the only approach that achieves improvements when training on noisy data with bouncing balls, as shown in Figure 5(red bars). In this training setup, it performs best on the standard test set without balls. Iso-Dream is built on a more efficient architecture than the baseline models. It provides a general framework that can be easily extended to other backbones.

**Qualitative results.** We visualize a sequence of predicted frames on BAIR with bouncing balls in Figure 6. Specifically, the output of two branches and corresponding masks are provided. We can see from these demonstrations that the world model of Iso-Dream is more accurate in modeling future dynamics for long-term prediction. It shows the fact that the action-free branch learns noncontrollable dynamics, while the action-conditioned branch learns controllable dynamics related to input action.

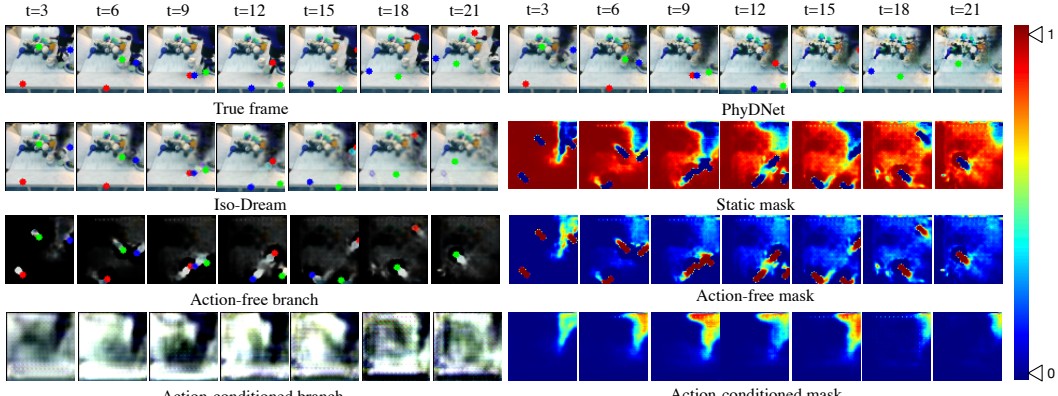

Figure 6: Showcases of video prediction results on the BAIR robot pushing dataset. We display every 3 frames in the prediction horizon. The generated masks show that each branch of Iso-Dream captures coarse localisation of controllable representations and noncontrollable representations.

## 5 Conclusions

In this paper, we proposed an MBRL framework named Iso-Dream, which mainly tackles the difficulty of vision-based prediction and control in the presence of complex visual dynamics. Our approach has two novel contributions to world model representation learning and corresponding MBRL algorithms. First, it learns to decouple controllable and noncontrollable latent state transitions via modular network structures and inverse dynamics. Further, it makes long-horizon decisions by rolling-out the noncontrollable dynamics into the future and learning their influences on current behavior. Iso-Dream achieves competitive results on the CARLA autonomous driving task, where other vehicles can be naturally viewed as noncontrollable components, indicating that with the help of decoupled latent states, the agent can make more forward-looking decisions by previewing possible future states in the action-free network branch. Besides, Iso-Dream was shown to effectively improve the visual control task in a modified DeepMind Control Suite, as well as the visual prediction task on the BAIR robot pushing dataset and the RoboNet dataset.

One limitation of Iso-Dream is the computational efficiency. Compared with DreamerV2, it requires longer training time per episode due to more intensive state transitions in behavior learning. But fortunately, from Figure 4(a), Iso-Dream is more sample-efficient than existing MBRL methods. Another limitation is the special treatment for different environments. In our preliminary experiments, we attempted to use the same model architecture for all test benchmarks. However, we observed that different benchmarks have specific requirements on the network structure, which we found should be dependent on our prior knowledge of the environments.

## Acknowledgements

This work was supported by the Natural Science Foundation of China (U19B2035, 62106144), Shanghai Municipal Science and Technology Major Project (2021SHZDZX0102), and Shanghai Sailing Program (21Z510202133).

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
