# OpenReview forum: "Iso-Dream: Isolating and Leveraging Noncontrollable Visual Dynamics in World Models"
_NeurIPS.cc/2022/Conference — NeurIPS 2022 Accept_

### Official Review · Reviewer_CMPp · 2022-06-23

**Rating:** 7
**Confidence:** 5
**Soundness:** 3 good
**Presentation:** 3 good
**Contribution:** 3 good

**Summary:**

This paper expands on the Dreamer framework to explicitly model controllable vs non-controllable dynamics, leveraging the fact that controllable dynamics can be explained away using an inverse dynamics model, while non-controllable dynamics ought to be predictable with no action change.

**Questions:**

- Have the authors looked into how the factorization deals with 'partially controllable' dynamics? When driving, other cars have their own (uncontrollable) intents, but are also affected by one's driving decisions. I suspect the model would still largely treat them as entirely uncontrollable due to the assumption that inverse dynamics ought to be able to resolve them if they were controllable, but I don't have a strong intuition for the actual outcome. The question may be relevant to leveraging this model to build models of other agents in the scene, besides the main one, and using those for behavior prediction.
- I understand the used of GRUs and LSTMs follows previous papers on the topic, but since everyone seems to have moved on to Transformer-based architectures, it begs the question of whether replacing the backbones with that kind of architecture would further improve the model's efficiency?
- I may have missed this from the discussion, but is it well-established that the background branch is necessary now that controllable and non-controllable effects are modeled separately? I didn't see an ablation for that.


**Limitations:**

Limitations section is terse and is a non-limitation in disguise.

**Strengths And Weaknesses:**

Strengths:
- strong academic contribution: the approach is novel, well-motivated, and well supported by evidence.
- the problem of modeling controllable vs. non-controllable effects is at the center of the problem of learning visual dynamics model, and this general area of research is highly relevant to a number of robotics applications.

Weaknesses:
- the limitations section is short. It would have been interesting to expand on things attempted but that didn't work for instance.

---

> ### Author Response · Authors · 2022-08-02
> **Our response to Reviewer CMPp**
>
> We thank the reviewer for the encouraging and constructive comments.
>
> **Q1. The limitations section is short. It would have been interesting to expand on things attempted but that didn't work for instance.**
>
> Thanks for the suggestion. In preliminary experiments, we attempted to use the same model architecture for all test benchmarks. However, the experimental results show that different benchmarks have specific requirements on the network structure, which we found should be dependent on our prior knowledge of the environments. For example:
> - For video prediction, we adopt PredRNN-V2 as the backbone network. We found that using the RSSM from Dreamer-V2 would decrease the quality of generated future images. It is because the compact states and a rather simple state transition model (in the form of a single-layer GRU) may inevitably result in information loss of video details.
> - For the control tasks, in DMC, we only disentangle the non-controllable states from the mixed visual data, but do not roll out them into the future and use them for behavior learning, as we do in the CARLA environment. We believe that whether the non-controllable dynamics should be considered as noise or as the condition of behavior learning is related to the specific settings of the environment. Especially in a virtual environment like DMC, it is unreasonable to force the model to learn the influence of background noise on the control strategy.
>
> We include the above discussion in the revision. It is worth noting that while we use specific network structures in different environments, Iso-Dream follows a unified disentanglement learning framework, which greatly benefits downstream visual prediction or control tasks.
>
> **Q2. Have the authors looked into the factorization deals with 'partially controllable' dynamics? ...leveraging this model to build models of other agents in the scene.**
>
> (1) Indeed, learning to separate 'partially controllable' dynamics can enable the agent to make decisions by anticipating the environmental reactions in advance, and is a promising research direction of model-based reinforcement learning (MBRL).
>
> However, in **Visual MBRL**, it is also a challenging topic, because it remains an open problem how to perfectly disentangle controllable and non-controllable dynamics from high-dimensional observations, let alone the factorization of visual dynamics into three terms. Therefore, in this paper, we mainly focus on the entire pipeline of disentangling the world model in two parts and exploiting the isolated future dynamics for decision-making.
>
> (2) We would like to make a clarification in response to the reviewer's comment that '*I suspect the model would still largely treat them as entirely uncontrollable due to the assumption that inverse dynamics ought to be able to resolve them if they were controllable.*'
>
> In CARLA, all pixel values in the entire image will change as the camera on the main car (*i.e.*, the agent) moves, so that we can view the dynamics of other vehicles as a combination of controllable and non-controllable states. Notably, in Fig. 3 (right) in the main manuscript, while the 'action-free masks' present hot spots around other vehicles, the pixel intensities in corresponding areas on the 'action-cond masks' are still greater than zero.
>
> **Q3. Transformer-based architectures instead of RNN-based architectures.**
>
> Thanks for the suggestion. First, in the world model, we use recurrent networks to learn the prior state transitions, $s_{t+1} \sim p(s_{t+1} | s_t, a_t)$ and $z_{t+1} \sim p(z_{t+1} | z_t)$, which focuses more on the temporal dependencies given the current state with/without a certain action. In this scenario, it can be inappropriate to use the self-attention mechanism of the Transformer to establish strong connections between the next state and a long sequence of previous states.
>
> Second, it is worth noting that in the policy network, we use similar attention mechanisms as in Transformer to adaptively incorporate the controllable state with multiple steps of non-controllable states.
>
> **Q4. The necessity of the background branch.**
>
> As shown in Fig. 4(b) in the revision, we have added an ablation study of time-invariant modeling. The performance of Iso-Dream decreases by about 15% in the absence of a separate network branch that captures the static information.

---

> > ### Comment · Reviewer_CMPp · 2022-08-03
> > **Thank you for the rebuttal.**
> >
> >
> > This makes perfect sense, and in combination with the feedback and responses from other reviewers this increases my confidence to a 5.

---

### Official Review · Reviewer_CjTg · 2022-07-11

**Rating:** 7
**Confidence:** 4
**Soundness:** 3 good
**Presentation:** 3 good
**Contribution:** 3 good

**Summary:**

This paper presents Iso-dream, which uses a variant of RSSM as a world model that explicitly represents controllable and non-controllable aspects of the world state.  Because these factors are modeled independently, Iso-dream allows its policy to condition on samples of future non-controlled state rollouts from the model.  This method is experimentally evaluated on several benchmarks, some to evaluate the policy and world model together, and some to evaluate the performance of dynamics predictions.  In all benchmarks, Iso-dream outperforms examined baselines.

##Update after responses

I'd like to thank the authors for the clear responses, both through discussion and experimentation, to my questions and concerns.  I have raised my score.

**Questions:**

Some issues to help with my understanding, or for the authors to address:
- Comparison to prior work as mentioned above
- In the CARLA environment, agent actions have the potential to control all pixels (turning, or accelerating, would change the positions of other cars in the observations as well as the background).  Can you clarify what controllable and non-controllable aspects of the state would be in this scenario?  What training signal makes the non-controllable mask represent other cars, and is that signal brittle?
- When rolling out the state for conditioning the policy, is there a reason this cannot be done with controllable state as well?  I wonder how a baseline that used undifferentiated world state like RSSM, but conditioned its policy on imagined future rollouts would perform (and also how well Iso-dream would work without this future rollout conditioning)  I am curious how much the controllable vs non-controllable split contributes to the improvement in results compared to the conditioning and would love to see these additional experiments in the ablations in Fig 4b.  Since I see this as the main contribution of the paper, I'd like to see this explored a bit more.
- For the BAIR and RoboNet datasets, I'm curious whether the improved results for Iso-dream are attributable more to the action-free or action-dependent parts of the input.  Is Iso-dream doing better reconstructing the 'distractions' than other approaches but performing similarly for the 'relevant' parts of the observations?

My score depends mostly on the fact that I think this is a solid, incremental contribution that is well evaluated but could be better.  My rating would be higher if my questions either result from some misunderstanding on my part or can be addressed in a response.

**Limitations:**

Computational limitations are mentioned, societal impact is not discussed.

**Strengths And Weaknesses:**

### Originality
The approach, to the best of my knowledge, is novel.  Some aspects of the separability of controllable and non-controllable states have been explored (InfoPower, as cited in the paper), and use of inverse dynamics losses are not new on their own.  However, the conditioning of the policy on future rollouts of the model is an interesting idea that I have not seen in this form before.  It is somewhat reminiscent of methods that incorporate planning into policy learning, and I would like to see the authors discuss the differences between this aspect of their approach and work like [Wang and Ba, Exploring Model-based Planning with Policy Networks, ICLR 2020].

### Quality & Clarity
The paper is well-written, technically sound, and mostly clear.  The diagrams are informative and help scaffold a quick understanding of Iso-dream. I'll list a few suggestions and questions that may aid in clarifying a few points, or which may be typos
- In line 130, the paper mentions a training collapse, but this collapse is not described
- Figure 5 contains a label that reads 'Truth futher'
- The image results in Figure 3 are somewhat small and difficult to make out

### Significance
The changes proposed in Iso-dream are incremental improvements to both the world model and policy learning of a prominent model-based RL approach.  These changes are evaluated in a thorough set of experiments.  This paper is a solid contribution.

---

> ### Author Response · Authors · 2022-08-02
> **Our response to Reviewer CjTg (1/2)**
>
> We thank the reviewer for the constructive comments.
>
> **Q1. Comparison to prior work [Wang & Ba, 2020].**
>
> Thank you for the suggested related work. We summarize the similarities and differences below:
>
> | Sim. / Diff.  | Iso-Dream | POPLIN    |
> | ---- | ---- | ---- |
> | Visual input     | Yes | No  |
> | Dynamics modeling      | Yes    | Yes    |
> | Policy network | $\pi\_\theta(\tilde{s}\_t, \tilde{z}\_{t:t+\tau})$ | $\pi\_\theta(s\_t)$   |
> | Behavior learning      | On-policy actor-critic learning on latent imaginations   | Policy distillation (*e.g.*, Behavior cloning) via off-policy experience replay     |
> | Planning algorithm     | n/a (only rolling out $\tilde z\_{t:t+\tau}$ but no actions)    | CEM planning in the action space by rolling out $\\{s\_{t:t+\tau}, a\_{t:t+\tau}\\}\_{1:K}$ |
> | Act in the environment | Directly execute the signal produced by the policy network given current observation | MPC: Run the planning algorithm and only execute the first action   |
>
> Notably, our approach is significantly different in five aspects:
> - Input data: Our approach is performed on high-dimensional visual observations and therefore deals with more challenging tasks.
> - Dynamics modeling: In Iso-Dream, the dynamics model backpropagates the gradients of value estimation to optimize sequential decision-making, while the learned dynamics in POPLIN only serve for computing the multi-step rewards for planning.
> - The conditioning of the policy network: Iso-Dream optimizes $\pi\_\theta(\tilde{s}\_t, \tilde{z}\_{t:t+\tau})$, which incorporates current state and future non-controllable states through an attention mechanism for more proactive decision-making. By contrast, POPLIN optimizes $\pi\_\theta(s\_t)$, which doesn't consider future changes of the non-controllable parts in the environment.
> - Behavior learning: In Iso-Dream, we learn the world model so that the behavior of the agent can be optimized on this simulator in an on-policy manner but without further interactions with the real environment. In POPLIN, the policy is simply learned on the replay data buffer through 'policy distillation' (*e.g.*, behavior cloning).
> - Act in the environment: Unlike POPLIN, our approach does not require a policy search process that generates a large number of trajectories $\\{s\_{t:t+\tau}, a\_{t:t+\tau}\\}\_{1:K}$, and therefore presents a more efficient way to interact with the environment from this perspective. But it can still be seamlessly integrated with typical planning algorithms such as random shooting or cross-entropy methods.
>
> **Q2. Description of 'training collapse' in Line 130.**
>
> Training collapse refers to the fact that during disentanglement learning, the action-conditioned branch captures most of the useful information, while the action-free branch learns almost nothing. This phenomenon may occur if the non-controllable component of the environment is simple, such that in Eq. (4), the action $a\_{t-1}$ can be easily predicted from the entangled state representations, without isolating the controllable dynamics.
>
> To avoid the training collapse, in Eq. (5), we use the prior state $\tilde{s}\_t$ in the action-conditioned branch, while using the posterior state $z\_t$ in the action-free branch. Intuitively, we want to increase the importance of the action-free branch in image reconstruction, so that $z\_t$ can encode informative representations rather than the 'white noise' in the environment.
>
> **Q3. About Fig. 3 and Fig. 5.**
>
> Thanks for pointing out these problems. We've fixed them in the revision.
>
> **Q4. Controllable and non-controllable components in CARLA.**
>
> In CARLA, we agree that the agent actions potentially affect all pixel values in the observation, as the camera on the main car (*i.e.*, the agent) moves. Therefore, we can view the dynamics of other vehicles as a combination of controllable and non-controllable states.
>
> Accordingly, our model can determine which component is dominant by learning attention masks (values between 0 and 1) across the action-conditioned and action-free branches. We draw these conclusions from Fig. 3 (right) in the manuscript, where the 'action-free masks' present hot spots around other vehicles, while the attention values in corresponding areas on the 'action-cond masks' are still greater than zero.
>
> We include the above discussion in the revision. Besides, in Section 4.3, we rephrase the relationship between the visual dynamics of other vehicles in CARLA and the two state components learned by the model.

---

> > ### Author Response · Authors · 2022-08-02
> > **Our response to Reviewer CjTg (2/2)**
> >
> > **Q5. (1) When rolling out the state for conditioning the policy, is there a reason this cannot be done with controllable state as well? (2) How well Iso-dream would work without this future rollout conditioning?**
> >
> > (1) In our setup (that we use only learn one policy network), current actions can only be derived from the current states and the imagined non-controllable states in the future, such that $a\_t \sim \pi(a\_t | \tilde{s}\_t, \tilde{z}\_{t:t+\tau})$, in which the inputs are not conditioned on $a\_t$ itself. It is not possible to have $a\_t \sim \pi(a\_t | \tilde{s}\_{t:t+\tau}, \tilde{z}\_{t:t+\tau})$, as the controllable states $\tilde{s}\_{t+1:t+\tau}$ are in turn dependent on $a\_{t:t+\tau-1}$. Therefore, this formulation of $\pi$ violates the causality of the Markov decision process. It shows the beauty of our approach that the separation of controllable and non-controllable states makes it possible to foresee $\tilde{z}\_{t:t+\tau}$ before executing $a\_t$ in the environment.
> >
> > We here propose a compromise solution to condition the policy on future rollouts of the controllable states. Specifically, we may train two decision-making modules on the same agent, such that
> > - Policy $\pi\_\theta$: $a\_t \sim \pi\_\theta(a\_t | \tilde{s}\_{t:t+\tau}^\prime, \tilde{z}\_{t:t+\tau})$, which is the main policy.
> > - Policy $\pi\_\phi$: $a\_t^\prime \sim \pi\_\phi(a\_t^\prime | \tilde{s}\_t^\prime, \tilde{z}\_{t:t+\tau})$, which is used for providing the conditioning for $\pi\_\theta$.
> >
> > By executing $\pi\_\phi$, we can use $a\_{t:t+\tau-1}^\prime$ to collect an arbitrary number of future trajectories of the controllable states $\tilde{s}\_{t:t+\tau}^\prime$ before we perform the main policy $\pi\_\theta$. Note that $\pi\_\phi$ shares the same world model backbone as $\pi\_\theta$ that learns to factorize $s\_{t+1} \sim p(s\_{t+1} | s\_t, a\_t)$ and $z\_{t+1} \sim p(z\_{t+1} | z\_t)$, so that $\tilde{s}\_{t:t+\tau}^\prime$ and $\tilde{z}\_{t:t+\tau}^\prime$ are also disentangled. Therefore, in $\pi\_\phi$, we can also independently roll out the non-controllable states in advance.
> >
> > The above method is still a hypothesis, which we don't have enough time to implement. Compared with the original Iso-Dream, an obvious problem of the new method is that the computation overhead required for behavior learning may increase linearly with the number of the rolling-out trajectories of $\pi\_\phi$.
> >
> > (2) In Fig. 4(b), we show the results of Iso-Dream where the actions are drawn from current states without rolling out them into the future, *i.e.*, we have $a\_t \sim \pi(a\_t | \tilde{s}\_t, \tilde{z}\_{t})$ instead of $a\_t \sim \pi(a\_t | \tilde{s}\_t, \tilde{z}\_{t:t+\tau})$. In this figure, the baseline model, which we call 'Iso-Dream w/o rolling-out', is represented by the green curve. We can see that without future conditioning, the performance drops significantly, which validates the effectiveness of future rollouts based on disentangled states.
> >
> > **Q6. For BAIR and RoboNet, is Iso-dream doing better reconstructing the 'distractions' than other approaches but performing similarly for the 'relevant' parts of the observations?**
> >
> > Good question! According to our empirical study, Iso-Dream shows strong dynamics modeling capabilities for both the 'relevant' parts and the 'distractions'. Please refer to Fig. 3 in the supplementary materials, where we test all the compared models **on the clean data without distractions**. Below are the main results.
> >
> > - Train: Clean BAIR videos; Test: Clean BAIR videos.
> >
> > | Metric |  SVG  | SA-ConvLSTM | PhyDNet | Iso-Dream |
> > |:------ |:-----:| ----------- | ------- | --------- |
> > | PSNR   | 21.24 | 21.15       | 21.51   | 21.80     |
> > | SSIM   | 0.815 | 0.776       | 0.811   | 0.828     |
> >
> > - Train: Data with bouncing balls; Test: Clean BAIR videos.
> >
> > | Metric |  SVG  | SA-ConvLSTM | PhyDNet | Iso-Dream |
> > |:------ |:-----:| ----------- | ------- | --------- |
> > | PSNR   | 20.80 | 20.53       | 20.35   | 22.55     |
> > | SSIM   | 0.793 | 0.744       | 0.787   | 0.852     |
> >
> > The results demonstrate that, in both training setups (with/without distractions), Iso-Dream significantly outperforms other approaches in predicting the 'relevant' parts. More interestingly, it is the only one that gets improved when trained on the noisy data. Such a result validates the strong disentanglement capabilities of Iso-Dream, whereas other models are greatly influenced by the distribution shift between the training and test data, represented by the presence of the 'distractions'.

---

> > > ### Comment · Reviewer_CjTg · 2022-08-07
> > > **Picking nits**
> > >
> > > >Therefore, this formulation of $\pi$ violates the causality of the Markov decision process
> > >
> > > I don't think this is true, given that the states $\tilde{s}_{t+1:t+T}$ are not actually observed from the environment but instead sampled from the agent's model of the environment.  This does not take away from the current paper since this is speculation I asked for in my original review, and your hypothetical model in response is also an interesting one.  Thank you again for your responses.

---

> > > > ### Author Response · Authors · 2022-08-08
> > > > **Thanks again for your comments**
> > > >
> > > > Thank you for increasing your score!
> > > >
> > > > Indeed, $\tilde{s}\_{t+1:t+T}$ are sampled from the learned model instead of interactions with the environment.
> > > >
> > > > Here is why we claimed that drawing $a_t$ from $\pi(\tilde{s}\_{t:t+T}, \tilde{z}\_{t:t+T})$ would violate the causality of our approach (even in the imaginations of the world model): In $a_t \sim \pi(\tilde{s}\_{t:t+T}, \tilde{z}\_{t:t+T})$, the inputs $\tilde{s}\_{t:t+T}$ are drawn from $p(s\_{t+1} | \tilde{s}\_{t}, a_t), \ldots, p(s\_{t+T} | \tilde{s}\_{t+T-1}, a_{t+T-1})$, which means that $\tilde{s}_{t+1:t+T}$ and $a_t$ depend on each other mutually. As a result, the behavior learning process will fall into a **Chicken-and-egg** dilemma.
> > > >
> > > > Therefore, in the new hypothetical model, we propose to use an additional policy network $\pi\_\phi(\tilde{s}\_t, \tilde{z}\_{t:t+T})$ to plan $a\_{t:t+T-1}^\prime$ and draw $\tilde{s}\_{t:t+T}$ from $p(s_{t+1} | \tilde{s}\_{t}, a_t^\prime), \ldots, p(s\_{t+T} | \tilde{s}\_{t+T-1}, a\_{t+T-1}^\prime)$. It will break the above interdependent conditions.
> > > >
> > > > Thanks again for your valuable comments!

---

### Official Review · Reviewer_MP54 · 2022-07-11

**Rating:** 7
**Confidence:** 4
**Soundness:** 3 good
**Presentation:** 3 good
**Contribution:** 3 good

**Summary:**

This paper proposes a world model that can decompose controlled and uncontrolled factors in the environment. Concretely, their world model is composed of a stream that takes in actions as input and its state is encouraged to contain mostly controlled factors in the video by an inverse dynamics loss on the learned states; the stream that models uncontrolled factors does not take actions as input so that the environment is predicted purely on observations. The two streams as separated in the observation space by masks generated from each stream. In experiments the authors show that the proposed method outperforms other world models in a modified version of DMC and the CARLA driving environment. Finally, they also show the proposed method is able to perform action conditioned video prediction in the BAIR robot push and RobotNet datasets.

**Questions:**

Please see the weaknesses section.

**Limitations:**

There is no negative societal impact discussed.

**Strengths And Weaknesses:**

Strengths:
+ A world model that decomposes uncontrolled and controlled factors in the input observation.
+ Outperforms recent world models when the environment involves mostly controlled factors.


Weaknesses:
- Performance comparisons in similar battleground.
The authors clearly show an advantage of their model over the baselines in the setting the proposed model was designed for. For completeness, and to see whether the proposed model performs similar to the baselines, the authors may want to provide experiments in a setting where the baselines were tested. For example, the authors can just test their method on the non-modified DMC environment, and can provide results of this to show that proposed model is able to improve on the baselines on the settings in this paper, but doesn't make them much worse in the original settings the baselines were tested.


- Special treatment for different environments.
As far as I understand, the authors choose which version of the proposed world model to use depending on whether the uncontrolled dynamics serve a purpose for the environment (Lines 143-147). Should the model learn this without the need of being told/trained depending on the environment. To me, this takes from the generality of the model and could imply that the model is simply good in specific scenarios. Can the authors comment on this? Or provide results without special treatment?




- Video prediction setup, results and other baselines:
The video prediction setup in this paper is unusual. The authors deliberately put bouncing balls on top of the frames in order to show the proposed network has an advantage over other networks in the task of video prediction. I would suggest the authors directly test on the clean data and show the performance there. The model should be able to handle both the original and the modified version of the data.


There are also more up to date baselines missing from this comparison such as FitVid (https://arxiv.org/pdf/2106.13195.pdf)



Suggestions:

- Modeling relationships between current controlled state and future uncontrolled states.
In equation 7, the authors model relationships between the uncontrolled and controlled steps where the uncontrolled states are unrolled into the future to compute a representation that is used to make decisions on what action to take next. While this is sensible to provide the model a look into the future, the future also should consider what actions the agent may take. The uncontrolled environment also depends on what actions the controlled part of the environment take. For example, if a controlled car turns left, we should expect parts of the environment that are not seemingly controlled, to react to the left turn, and this may cause a chain reaction with all cars in the environment. Therefore, it seems more sensible to unroll both the controlled and uncontrolled states, and then equation 7 should be based on the result of this. If time permits, could the authors unroll both states together and see if this results in improvements? This is more of a suggestion out of curiosity and not a weakness of the model.

---

> ### Author Response · Authors · 2022-08-02
> **Our response to Reviewer MP54 (1/2)**
>
> **Q1. Experiments in a setting where the baselines were tested.**
>
> For clarification, please note that SVEA [25], the most competitive baseline model in our manuscript, was also tested on similar DMC settings with task-irrelevant video distractions in its paper. In fact, the environments we used were derived from the test benchmarks of SVEA, which ensures the fairness of the experimental comparison.
>
> Besides, per the reviewer's request, we further compare our approach with SVEA in non-modified DMC environments with a static and plain background. The results are as follows:
>
> | Task         | SVEA | DreamerV2 | Iso-Dream |
> |:------------ |:----:|:---------:|:---------:|
> | Walker Walk  | 882  |    763    |    924    |
> | Cheetah Run  | 252  |    705    |    589    |
> | Finger Spin  | 370  |    761    |    796    |
> | Hopper Stand | 172  |    781    |    826    |
>
> We hope that the above explanations and supplementary results can help the reviewer's concern about the DMC setup.
>
> **Q2. Special treatment for different environments.**
>
> In preliminary experiments, we attempted to use the same model architecture for all test benchmarks. However, the experimental results show that different benchmarks have specific requirements on the network structure, which we found should be dependent on our prior knowledge of the environments.
> - For video prediction, we adopt PredRNN-V2 as the backbone network. We found that using the RSSM from Dreamer-V2 would decrease the quality of generated future images. It is because the compact states and a rather simple state transition model (in the form of a single-layer GRU) may inevitably result in information loss of video details.
> - For the control tasks, in DMC, we only disentangle the non-controllable states from the mixed visual data, but do not roll out them into the future and use them for behavior learning, as we do in the CARLA environment. We believe that whether the non-controllable dynamics should be considered as noise or as the condition of behavior learning is related to the specific settings of the environment. Especially in a virtual environment like DMC, it is unreasonable to force the model to learn the influence of background noise on the control strategy.
>
> We include the above discussion in the revision as a limitation of our work. It is worth noting that while we use specific network structures in different environments, Iso-Dream follows a unified disentanglement learning framework, which greatly benefits downstream visual prediction or control tasks.
>
> **Q3. Video prediction setup, results, and other baselines.**
>
> (1) In Fig. 3 in the supplementary materials, we have shown the results of **directly testing on the clean data**. We use two training setups and summarize the main results as follows. In both setups, Iso-Dream significantly outperforms all of the compared models when tested on clean data.
>
> - Trained on clean BAIR videos (common setup).
>
> | Metric |  SVG  | SA-ConvLSTM | PhyDNet | Iso-Dream |
> |:------ |:-----:| ----------- | ------- | --------- |
> | PSNR   | 21.24 | 21.15       | 21.51   | 21.80     |
> | SSIM   | 0.815 | 0.776       | 0.811   | 0.828     |
>
> - Trained on data with bouncing balls.
>
> | Metric |  SVG  | SA-ConvLSTM | PhyDNet | Iso-Dream |
> |:------ |:-----:| ----------- | ------- | --------- |
> | PSNR   | 20.80 | 20.53       | 20.35   | 22.55     |
> | SSIM   | 0.793 | 0.744       | 0.787   | 0.852     |
>
> Notably, Iso-Dream is the only one that gets improved when trained on the noisy data. Such a result demonstrates that Iso-Dream has strong disentanglement learning capabilities, whereas other models are greatly influenced by the distribution shift between the training and test data, represented by the presence of the bouncing balls.
>
> (2) As suggested by the reviewer, we compare our model with FitVid [Babaeizadeh et al., 2021] for action-conditioned video prediction.  We train and test in the noisy data with bouncing balls, aligning the setting in our manuscript. As we can see from the results below, Iso-Dream consistently outperforms FitVid in both 18-frame and 28-frame prediction, and it shows a stronger capability of long-term prediction for complex data with mixed spatiotemporal dynamics.
>
> - Predict the next 18 frames
>
> | Model               | PSNR | SSIM | \# Params |
> |:------------------- |:----:| ---- | --------- |
> | FitVid              |   19.14   | 0.785 |   7.4M   |
> | Iso-Dream  |    21.43  |   0.832   |     8.1M      |
>
> - Predict the next 28 frames
>
> | Model               | PSNR | SSIM | \# Params |
> |:------------------- |:----:| ---- | --------- |
> | FitVid         |  17.60    |   0.739   |    7.4M    |
> | Iso-Dream  |    19.51  |  0.768 |       8.1M    |

---

> > ### Author Response · Authors · 2022-08-02
> > **Our response to Reviewer MP54 (2/2)**
> >
> > **Q4. (1) The uncontrolled environment also depends on what actions the controlled part of the environment takes. (2) Unroll both the controllable and non-controllable states together.**
> >
> > (1) Indeed, learning to separate 'partially controllable' dynamics (*i.e., the non-controllable parts will also react to what actions the controllable part of the environment takes*) can enable the agent to make decisions by anticipating the environmental reactions in advance, and is a promising research direction of model-based reinforcement learning (MBRL).
> >
> > However, in **Visual MBRL**, it is also a challenging topic, because it remains an open problem how to perfectly disentangle controllable and non-controllable dynamics from high-dimensional observations, let alone the factorization of visual dynamics into three terms. Therefore, in this paper, we mainly focus on the entire pipeline of disentangling the world model in two parts and exploiting the isolated future dynamics for decision-making.
> >
> > (2) Thanks for the suggestion. In our behavior learning algorithm, it is intractable to unroll the controllable states using a single decision-making module. The key that we can model the decision-making process as $a\_t \sim \pi(a\_t | \tilde{s}\_t, \tilde{z}\_{t:t+\tau})$ is that the input states are not conditioned on $a\_t$ itself. Note that it is the separation of controllable and non-controllable states that makes it possible to foresee $\tilde{z}\_{t:t+\tau}$ before executing $a\_t$ in the environment.
> >
> > In contrast, it is not possible to have $a\_t \sim \pi(a\_t | \tilde{s}\_{t:t+\tau}, \tilde{z}\_{t:t+\tau})$, as the controllable states $\tilde{s}\_{t+1:t+\tau}$ are in turn dependent on $a\_{t:t+\tau-1}$, which makes the formulation of $\pi$ violate the causality of the Markov decision process.
> >
> > **After careful consideration of the reviewer's suggestion, we here propose a compromise solution to condition the policy on future rollouts of the controllable states.** Specifically, we may train two decision-making modules on the same agent, such that
> > - Policy $\pi\_\theta$: $a\_t \sim \pi\_\theta(a\_t | \tilde{s}\_{t:t+\tau}^\prime, \tilde{z}\_{t:t+\tau})$, which is the main policy.
> > - Policy $\pi\_\phi$: $a\_t^\prime \sim \pi\_\phi(a\_t^\prime | \tilde{s}\_t^\prime, \tilde{z}\_{t:t+\tau})$, which is used for providing the conditioning for $\pi\_\theta$.
> >
> > By executing $\pi\_\phi$, we can use $a\_{t:t+\tau-1}^\prime$ to collect an arbitrary number of future trajectories of the controllable states $\tilde{s}\_{t:t+\tau}^\prime$ before we perform the main policy $\pi\_\theta$. Note that $\pi\_\phi$ shares the same world model backbone as $\pi\_\theta$ that learns to factorize $s\_{t+1} \sim p(s\_{t+1} | s\_t, a\_t)$ and $z\_{t+1} \sim p(z\_{t+1} | z\_t)$, so that $\tilde{s}\_{t:t+\tau}^\prime$ and $\tilde{z}\_{t:t+\tau}^\prime$ are also disentangled. Therefore, in $\pi\_\phi$, we can also independently roll out the non-controllable states in advance.
> >
> > The above method is still a hypothesis that we don't have enough time to implement. Compared with the original Iso-Dream, an obvious problem of the new method is that the computation overhead required for behavior learning may increase linearly with the number of the rolling-out trajectories of $\pi\_\phi$.

---

> > > ### Author Response · Authors · 2022-08-08
> > > **To Reviewer MP54**
> > >
> > > Dear Reviewer MP54,
> > >
> > > Thank you once again for your review of our work. As the discussion period is approaching its end, we would be grateful if you could confirm whether our responses and the additions we have made to the manuscript addressed your concerns. Please let us know if any issues remain.

---

> > > > ### Comment · Reviewer_MP54 · 2022-08-08
> > > > **Thank you**
> > > >
> > > > I would like to thank the authors for their efforts in the rebuttal. They have clarified all of my concerns. I am raising my score to accept. I do think it'll be interesting to try unrolling the policy as well as the environment in two separate streams that see each other. Hopefully this results into an interesting finding from the authors in the future.

---

### Author Response · Authors · 2022-08-02
**General Response: Revision Uploaded**

We thank all reviewers for their constructive comments and have updated our paper accordingly. Please check out the new version!

Specific changes include:

1. Clarify the so-called training collapse (Section 3.2).
2. Add an ablation study to show the benefits of using a separate network branch to model time-invariant information (Fig. 4(b)).
3. Give more discussion on the limitations of our work (Section 5).

Please don’t hesitate to let us know for any additional comments on the paper.

---

### Meta-Review · Area_Chair_Jh6i · 2022-08-20

**Recommendation:** Accept
**Confidence:** Certain

**Metareview:**

This paper studies the problem of building world models that can decouple controllable and uncontrollable factors in the environment. The paper received reviews that generally tended towards acceptance. However, the reviewers had difficulty understanding some details and had concerns that the setup might not be the same across environments. The authors provided a rebuttal that addressed most of the reviewers' concerns. The paper was discussed and all the reviewers updated their reviews in the post-rebuttal phase. Reviewers generally agree that the paper should be accepted. AC agrees with the reviewers and suggests acceptance. However, the authors are urged to look at reviewers' feedback and incorporate their comments into the camera-ready.

**Award:**

No

---

### Decision · Program_Chairs · 2022-09-14

Accept